# An Integrated Characterization Strategy on Board for Recycling of poly(vinyl butyral) (PVB) from Laminated Glass Wastes

**DOI:** 10.3390/polym16010010

**Published:** 2023-12-19

**Authors:** Vasilis Nikitakos, Athanasios D. Porfyris, Konstantinos Beltsios, Constantine Papaspyrides, Simone Bordignon, Michele R. Chierotti, Stefano Nejrotti, Matteo Bonomo, Claudia Barolo, Alessandro Piovano, Rudolf Pfaendner, Beatriz Yecora, Angelica Perez

**Affiliations:** 1Laboratory of Polymer Technology, School of Chemical Engineering, Zographou Campus, National Technical University of Athens, 15780 Athens, Greece; vnikitakos@mail.ntua.gr (V.N.); kgbelt@mail.ntua.gr (K.B.); 2NIS Interdepartmental Centre, Department of Chemistry, University of Torino, 10125 Torino, Italy; simone.bordignon@unito.it (S.B.); michele.chierotti@unito.it (M.R.C.); stefano.nejrotti@unito.it (S.N.); matteo.bonomo@unito.it (M.B.); claudia.barolo@unito.it (C.B.); 3National Reference Center for Electrochemical Energy Storage (GISEL)—INSTM, 50121 Firenze, Italy; alessandro_piovano@polito.it; 4GAME Lab, Department of Applied Science and Technology (DISAT), Politecnico di Torino, 10129 Torino, Italy; 5Fraunhofer Institute for Structural Durability and System Reliability LBF, 64289 Darmstadt, Germany; rudolf.pfaendner@lbf.fraunhofer.de; 6LUREDERRA Technological Centre, Perguita Industrial Area, 31210 Los Arcos, Spain; beatriz.yecora@lurederra.es (B.Y.); angelica.perez@lurederra.es (A.P.)

**Keywords:** polyvinyl butyral, laminated glass, recycling

## Abstract

Polyvinyl butyral (PVB) is widely used as an interlayer material in laminated glass applications, mainly in the automotive industry, but also for construction and photovoltaic applications. Post-consumed laminated glass is a waste that is mainly landfilled; nevertheless, it can be revalorized upon efficient separation and removal of adhered glass. PVB interlayers in laminated glass are always plasticized with a significant fraction in the 20–40% *w*/*w* range of plasticizer, and they are protected from the environment by two sheets of glass. In this work, the aim is to develop a thorough characterization strategy for PVB films. Neat reference PVB grades intended for interlayer use are compared with properly processed (delaminated) post-consumed PVB grades from the automotive and construction sectors. Methods are developed to open opportunities for recycling and reuse of the latter. The plasticizer content and chemical nature are determined by applying well-known analytical techniques, namely, FT-IR, TGA, NMR. The issue of potential aging during the life cycle of the original laminated material is also addressed through NMR. Based on the findings, a sensor capable of directly sorting PVB post-consumer materials will be developed and calibrated at a later stage.

## 1. Introduction

PVB (polyvinyl butyral) is a vinyl acetal polymer synthesized by the condensation reaction (called here acetalization) of polyvinyl alcohol (PVA) with butyraldehyde (BA) in the presence of an acid catalyst [1]. It demonstrates excellent mechanical strength, high-temperature resistance, strong adhesion to glass, UV stability, and optical transparency. In view of these features, PVB is extensively used for a range of applications, predominantly as a safety glass-supporting component [1,2]. Secondary uses of PVB pertain to sound insulation, photovoltaic modules, coatings, batteries, binders, adhesives, etc.

PVB presents a random terpolymer structure with segments mainly made of vinyl butyral units (up to 80%), 18–24% vinyl alcohol, and 1–4% vinyl acetate [1,3,4,5]. Backbone hydroxyl groups of vinyl alcohol segments can form hydrogen and covalent bonds to the surface of polar substrates and are largely responsible for the adhesion of the PVB interlayer–glass interface in the process of lamination [6]. PVB for lamination is prepared as a film by melt-mixing with plasticizers, since unplasticized vinyl acetal polymers are brittle, unpliable materials that are difficult to process. For that purpose, compatible plasticizers are incorporated into the polymeric matrix; the glass transition temperature (*T*_g_) of the polymer is lowered, and flexibility and ductility at ambient temperature are achieved. Phthalate-free plasticizers are used nowadays for the PVB plasticization, some of which are triethyleneglycol di-(2-ethyl hexanoate) (3GO) or tetraethylene glycol di-*n*-heptanoate (TEGH), dibutyl sebacate (DBS), dihexyl adipate (DHA), dioctyl adipate (DOA), hexyl cyclohexyl adipate, or mixtures of heptyl and nonyl adipates [7,8]. In the melt process, route plasticizers and PVB are premixed, and the resulting material is extruded in the melt state, while in the case of casting process routes, prior to casting, the plasticizer and PVB are initially dissolved in a common solvent [8,9,10]. 

Developments in the vinyl acetal polymer technology in the 1930s led to the patenting of PVB by the chemical company Union Carbide and Carbon Corporation in 1935 [10]. The application of polyvinyl butyral (PVB) had been primarily promoted by the car industry for safety glass (windshields) replacing cellulose nitrate. The traditional leading companies in PVB production are Eastman, Kuraray, Sekisui, Chang Chun Petrochemical (CCP). Additionally, some recently emerging companies, mainly in Asia, all produce thousands of tons per year of PVB film for automotive and construction use [4,11]. PVB holds the first place as an interlayer in safety glass, leaving behind substitutional options such as ethylene-vinyl acetate (EVA) or the ionoplast Sentry Glass Plus (SGP) [12]. The global PVB films market is projected to grow considerably in the coming years due to increasing industrialization and urbanization, with the market size reaching USD 3641.25 million by 2031, progressing at a compound annual growth rate (CAGR) of 3.6% during the forecast period [11]. Worldwide, 65% of all PVB films are used in automotive applications and as windshields, containing approximately 1 kg of PVB per 13 kg of laminated glass. There is no end-of-life cycle program for ca. 90 of PVB film per year [13], and the latter material ends up in landfills. Recycling of PVB is practically important as the price of the material ranges from $6.50 to $9.50 per kg depending on its grade in the United States and Europe [14,15]. Additionally, PVB film might contain approximately 30% *w*/*w* plasticizers, which are also valuable and are being wasted. Therefore, there is a need to recycle a valuable waste stream such as PVB.

The main issues for the recycling of laminated glass are the separation of the glass from the polymer, the categorization of the different PVB samples and the potential degradation incurred during the life cycle of PVB or the reprocessing step [10]. Up to now, most of the post-consumed PVB interlayer material is incinerated or landfilled, and only 9% is recycled for secondary uses [16]. Based on scientific publications and existing patents in this field [8,13,17,18], the delamination process includes mechanical treatment by either shattering the glass or separating the glass, followed by chemical separation involving acids or bases in the presence of heating and, finally, cleaning, washing, and drying. 

Retrieved PVB should also be sorted into categories (e.g., plasticizer type and/or content, extent of degradation) for better management and optimum recycling. In this context, an innovation of the present work is the integrated characterization strategy proposed as a preliminary step in the recycling of PVB. Identifying the different grades of PVB and the most commonly used plasticizers will enable the sorting of laminated glass for PVB recycling. Novel strategies based on the combination of several advanced analytical techniques and the application of spectral data are proposed to provide a more complete evaluation of the composition and degradation level of the laminated layer. 

Based on this holistic approach, in the current study, part of a strategy for high-quality recycling and valorization of PVB from laminated glass into new interlayer material is demonstrated. We aim to set the basis for the categorization of various PVB grades and films by characterizing off-line different commercial PVB reference materials by applying several methods, namely FT-IR (Fourier-transform Infrared spectroscopy), solution and solid-state NMR (Nuclear Magnetic Resonance), TGA (Thermogravimetric Analysis), DSC (Differential Scanning Calorimetry), MFR (Melt Flow Rate), and DSV (dilute solution viscometry). To the best of our knowledge, the combination of different techniques has not been applied to develop an accurate, in-line, rapid, and robust inspection solution for the characterization of the composition and quality of PVB films in laminated glass. Benchmarking of the critical properties of reference PVB samples will allow determining the optimum level of properties that a post-consumed grade should meet, in order to be reused as an interlayer material (“closed loop”).

## 2. Materials and Methods

In order to perform the analysis of the different films for glass lamination, a set of commercial references from Eastman (PVBR- samples) were analyzed. The PVBR- series was selected because they contain different plasticizers, while an unplasticized PVB grade was used for comparison. Moreover, two different types of laminated glass waste from automotive (glass waste automotive, GWA) and construction (glass waste construction, GWC) were used. Lastly, PVB-GR-AUTOMOTIVE was provided as a reference for a new windscreen for comparison purposes in the NMR analysis. The extraction of PVB from the GWA and GWC waste samples and from the new windscreen was achieved through a special mechanochemical treatment process developed by Lurederra Technological Centre [16]. Accordingly, the glass waste samples undergo mechanical shredding using a hammer mill and a blade mill, resulting in a significant portion of the glass being transformed into a powdered form. This effectively facilitates the separation of PVB from the glass. Subsequently, a chemical treatment was conducted, involving a sequential series of five steps of washing with acidic and alkaline reagents. The acidic solutions are used to weaken the existing bonds between glass fragments and PVB, while the alkaline solution serves to neutralize residual acids and eliminate impurities. Details about the analyzed samples can be found in Table 1

### 2.1. Fourier Transform-Infrared Spectroscopy (FT-IR)

The ATR (Attenuated Total Reflection) FT-IR spectroscopy is the most common method for obtaining IR spectra of thin films/plates or liquids. Measurements were performed on a Bruker AII ATR Spectrometer (Bruker Corporation, Billerica, MA, USA) from 400 to 4000 cm^−1^ with a 4 cm^−1^ resolution, using a diamond crystal. Thirty-two co-added spectra were taken for each measurement. FT-IR was also used to develop a master curve to directly determine the plasticizer content. All samples characterized via FT-IR were in the form of films. 

### 2.2. Thermogravimetric Analysis (TGA)

The thermal stability of the PVB samples was assessed by thermogravimetric analysis (TGA) in a Mettler Toledo TGA/DSC HT 1 apparatus (Mettler Toledo, Columbus, OH, USA). The samples were analyzed through dynamic heating from 30 to 600 °C under controlled nitrogen flow (10 mL min^−1^) at a heating rate of 10 °C min^−1^. The onset of decomposition was defined as the temperature at 5% weight loss (*T*_5%_), the degradation temperature (*T*_d_) was determined at the maximum rate of weight loss (1st derivative curve), and the char yield as the wt.% residue left at 600 °C [19]. In addition, isothermal TGA runs were also conducted at 250 °C for 2 h to completely and selectively evaporate the plasticizer, leaving the polymer practically intact. Measurements were performed in duplicate for all received PVB samples so as to check the reproducibility of the samples. All samples characterized via TGA analysis were in the form of films.

### 2.3. Differential Scanning Calorimetry (DSC)

Differential Scanning Calorimetry (DSC) measurements were conducted in a Mettler DSC 1 module (Mettler Toledo, Columbus, OH, USA). A heating–cooling–heating cycle from −20 to 210 °C at a heating (cooling) rate of 10 (−10) °C min^−1^ was employed to erase the thermal history of the material and impose identical cooling/crystallization conditions between the different samples [18]. Nitrogen flow was controlled at 20 mL/min. From the received curves the glass transition temperature (*T*_g_) and the melting point (*T*_m_, if any) of the PVB samples were determined. Samples measured by DSC were in the form of films.

### 2.4. Solution and Solid-State Nuclear Magnetic Resonance (NMR) 

For solution measurements, a JEOL 600 ECRZ instrument (JEOL Ltd, Akishima, Tokyo, Japan) operating at 600 MHz for ^1^H was used. For each measurement, 20 mg of sample were firstly dispersed in deuterated methanol (1 mL) and then heated (40 °C) and sonicated to allow for a complete dissolution. In order to obtain reliable and fully quantitative results, a dedicated NMR procedure was set up (i.e., 64 scans, relaxation delay of 10 s). Aiming at the quantification of the relative amounts of polymer’s monomeric unit (e.g., the vinyl butyrate and vinyl alcohol), we relied on the approach developed by Corroyer et al. [4].

The SSNMR (Solution and Solid-State Nuclear Magnetic Resonance) spectra were acquired with a Bruker Avance II 400 Ultra Shield instrument (Bruker Corporation, Billerica, MA, USA), operating at 400.23 and 100.63 MHz for ^1^H and ^13^C nuclei, respectively. The powdered samples were packed into a cylindrical zirconia rotor with a 4 mm o.d. and an 80 μL volume and spun at 8 kHz. A certain amount of sample was collected from each batch and used without further preparations to fill the rotor. The ^13^C CPMAS (Cross Polarization Magic-Angle Spinning) spectra were acquired at room temperature using a ramp cross-polarization pulse sequence with a 90° ^1^H pulse of 3.60 μs, a contact time of 50 μs, a recycle delay of 1 s, and a number of scans of 2000. Direct excitation ^13^C MAS spectra were acquired with a 90° ^13^C pulse of 4 μs, a recycle delay of 1 s and 1400 scans. For both CPMAS and MAS experiments, a two-pulse phase modulation (TPPM) decoupling scheme was used, with a radiofrequency field of 69.4 kHz. The ^13^C chemical shift scale was calibrated through the methylenic signal of external standard α-glycine (at 43.7 ppm). All PVB samples characterized by NMR were in the form of powders. PVB samples were pulverized by cryo-milling (with liquid nitrogen) in a Fritsch pulverisette apparatus (FRITSCH Milling and Sizing, Inc. 27312 Pittsboro, NC, USA), so as to facilitate the dissolution of PVB. 

### 2.5. Dilute Solution Viscometry 

Dilute Solution Viscometry (DSV) was used to define the intrinsic viscosity (IV, [*η*]) of the PVB samples. Dilute solutions of 0.5 g·dL^−1^ concentration of each PVB sample in tetrahydrofuran (THF) were prepared. Measurements were performed in an Ubbelohde-type viscometer (Cannon Instrument Company, State College, PA, USA) (K = 0.009340 mm^2^·s^−2^) at 25 ± 0.1 °C. The outflow times of THF and each PVB solution were measured at least in triplicate and the [*η*] values were obtained by single-point measurement, via Equation (1) [20].
(1)$$[\eta ]=\frac{\sqrt{1+1.5\eta_{sp}}+1}{0.75C}$$

Furthermore, especially in the case of the PVB/THF system, the viscosity average molecular weight (${\bar{M}}_{\nu }$) could be extracted from the Mark–Houwink–Sakurada equation (Equation (2)), by using the empirical parameters, K = 2.52 × 10^−4^ mL/g and a = 0.72 as suggested in the literature, for PVB with vinyl alcohol (VA) content close to 10% [21].
(2)$$[\eta ]=K{\left({\bar{M}}_{\nu }\right)}^{a}$$

### 2.6. Gel Permeation Chromatography (GPC)

GPC was carried out with the use of Agilent 1260 Infinity II instrument (Agilent Technologies, Santa Clara, CA, USA), equipped with a guard column (PLgel 5 μm) and two PLgel MIXED-D 5 μm columns (300 × 7.5 mm). Elution was carried out with THF (Tetrahydrofuran) (≥99.8% purity, Fisher Scientific UK Ltd., Loughborough, UK) at a flow rate of 1 mL⋅min^−1^. The analysis was performed using an Agilent 1260 Infinity II refractive index detector (RID) (G7162A), (Agilent Technologies, Santa Clara, CA, USA). The calibration of the instrument was carried out with polystyrene standards of molecular weight from 162 to 500,000 g·mol^−1^ (EasiVial PS-M 2 mL, Scientific Laboratory Supplies Ltd. UK, Wilford, Nottingham, UK), and a universal calibration curve was constructed.

### 2.7. Soxhlet Extraction (SE)

The removal of plasticizers from the PVBR series of samples was attempted through the application of the Soxhlet Extraction (SE) process. SE is an extraction technique applied to analytes that are sufficiently thermally stable [22,23,24]. The extraction solvent is continuously cycled through the matrix by boiling and condensation, with the sample being at a certain position and the solvent circulating through it [25,26]. The duration of the extraction was set at 12 h, and the solvent used was hexane. The PVBR- films were milled down to a thin powder (with the use of liquid nitrogen) to increase the surface area of the samples. 5 g of each sample was added to a typical Soxhlet paper filter (thimble), and the sample plus paper filter were pre-weighed. After 12 h of extraction, the paper filter was removed from the apparatus, and it was left to dry overnight in the fume hood, followed by another 4 h drying under vacuum at 80 °C, so as to totally remove the solvent. The paper filter with the remaining sample after extraction was weighed after drying and compared to the initial weight, to close the mass balance. Furthermore, the PVB samples after extraction (PVBR-SE) were compared to the respective reference samples by TGA analysis to gauge the amount of plasticizer removed from the samples.

### 2.8. Melt Flow Rate (MFR)

Melt Flow Rate (MFR) measurements were taken for the neat PVB (PVB APS) and all the plasticized PVB grades (PVBR-B, E, F and waste samples) prepared according to standard ISO 1133 [27]. The measurements were performed at NTUA in a Kayeness Dynisco 4004 rheometer (Dynisco Europe GmbH, Heilbronn, Germany) at 190 °C, with a load of 10 kg for the unplasticized grade (PVB APS) and 2.16 kg for all the plasticized samples. All samples were dried at 60 °C under vacuum for 6 h prior to each measurement, since PVB is mildly hygroscopic [25,28].

## 3. Results and Discussion

### 3.1. Plasticizer Type 

Polyvinyl butyral is a terpolymer that consists of three main segments, namely Vinyl Butyral (VB, 76–80 wt%), Vinyl Alcohol (VA, 18–22 wt%) and Vinyl Acetate (VAc, 1–2 wt%) (Figure 1) [10]. 

The acetalization degree (vinyl butyral, VB content) is related to the manufacturing process, meaning that the chemical composition (frequency of X, Y, and Z segments, Figure 1) can differ among different PVB grades and suppliers, thus affecting the final properties of the end product [29]. Nonetheless, PVB is found to be mainly composed of polyacetal, since it contains a predominant proportion of butyral, and much less hydroxyl and acetyl groups [4]. FT-IR provides the means to identify the characteristic peaks of PVB moieties but without the possibility of a quantitative analysis of the chemical composition. For the neat PVB sample (PVB APS) (Figure 2a), the peaks at about 1140 and 996 cm^−1^ correspond to C–O–C stretching vibrations of butyraldehyde groups [26]. A broad peak related to hydroxyl content (3490 cm^−1^) is visible in the spectra of all PVB samples referring to vinyl alcohol (VA) content, while the vinyl acetate (VAc) part of PVB (1740 cm^−1^) is almost indistinguishable for PVB APS due its low content (assumed <1%), which is close to the lower detection limit of the FT-IR technique. PVB films used for lamination also contain a high amount of plasticizer (15–45 wt.%), leading to a lowered *T*_g_ value and enhanced film flexibility. Plasticizers used in PVB films for lamination typically involve ethylene glycol oligo-esters, or esters of dicarboxylic acids like dibutyl sebacate [10,23,25,30]. As for the spectra of pure 3GO and DBS plasticizers (Figure 2), they exhibit different peaks than PVB in the range of 1000–1500 cm^−1^, which are attributed to their different chemical structures (Figure 1), but they cannot be easily determined when mixed with PVB, due to the overlapping of characteristic vibrational modes. The most characteristic peak is found at 1740 cm^−1^ for both plasticizers, representing the C=O stretching, which is indicative of an ester moiety [30]. Regarding the FT-IR analysis of the plasticized PVB grades (Figure 2a), the peak assignment involves characteristic peaks of the PVB terpolymer (VA, VB, VAc units), as well as one at ca. 1740 cm^−1^, proving the presence of the plasticizer. The particular peak is not observed in the spectrum of the unplasticized sample (PVB APS). Nevertheless, according to FT-IR, it is feasible to determine the plasticizer type (e.g., ester type) but not the exact species, 3GO or DBS. All the spectra were normalized for qualitative comparison purposes in order to assess the influence of plasticizer because carbonyl groups are associated with the presence of plasticizer molecules.

Regarding the FT-IR spectra analysis of GW samples (Figure 2b), there is no clear difference from the reference samples (Figure 2a). To clarify that, PVB waste grades were compared to PVBR-B (Figure 2c), and all of them contain 3GO as a plasticizer (as shown by NMR analysis below). An overlapping of the characteristic peaks and minor differentiation in the intensity of the peaks were only observed, proving there is no great chemical variation between PVB films after their end-of-life. Therefore FT-IR cannot be used for the study of aging or degradation phenomena of this type and extent that are typical for the waste materials of practical interest.

SSNMR is another technique that was applied in the characterization of the PVBR- samples. Thanks to the variety of available experiments, it can easily be used for plastic materials, investigating the more rigid fraction (represented by the PVB polymer) or the dynamic fraction (consisting of the plasticizer). This can be promptly achieved by acquiring ^13^C CPMAS with short contact times or ^13^C MAS, respectively.

First, PVBR-B was compared to a reference sample from a new windshield abbreviated as PVB-GR-AUTOMOTIVE (Figure 3). The two materials coincide, both in their rigid (PVB) and mobile (plasticizer) domains. 

In the CPMAS spectra (top in Figure 3), broad signals can be observed, corresponding to the C atoms of the main component of PVB (i.e., VB). As for the MAS spectra (bottom in Figure 3), the signals appear much narrower, as they correspond to the C atoms of mobile 3GO. Peak assignments both for PVB and 3GO are proposed in the form of labels over the corresponding resonances in Figure 3.

PVBR-B was further compared with samples PVBR-E and PVBR-F. Figure 4 shows an overlay of their ^13^C CPMAS (top) and MAS (bottom) SSNMR spectra.

As shown by the spectra (Figure 4 on top), no significant difference among the three samples is found in their rigid polymeric part. On the contrary, as expected, several evident variations are observed concerning the plasticizers included in the three samples. The details of colored arrows in Figure 5 make it easier to see how specific signals can be identified in the three spectra that allow for the discrimination of the incorporated plasticizer.

PVBR-B, which is characterized by the presence of the 3GO plasticizer, is the sample whose spectrum differentiates itself the most. Many of the visible signals (e.g., those at 70.9, 69.6, 47.4, and 12.2 ppm) are well separated from those of DHA (in PVBR-E) and DBS (in PVBR-F). Nonetheless, the latter two plasticizers also offer certain specific diagnostic signals: DHA displays a characteristic couple of resonances at 26.3 and 25.1 ppm, while the peaks at 31.5 and 19.8 ppm appear to be uniquely ascribable to DBS. This shows how SSNMR has the potential of reliably assessing the chemical identity of the plasticizer included in the analyzed materials.

Regarding GW samples, Figure 6 shows the ^13^C SSNMR spectra of the three analyzed batches of PVB, namely PVB-GW-AA-01, PVB-GW-C-01, and PVB-GW-NA-04.

Once again, if the signals referring to the polymer component of the materials (top in Figure 6) are superposable, those ascribable to plasticizers are not; indeed, while PVB-GW-C-01 and PVB-GW-NA-04 are completely coincident, both exhibiting only signals relative to 3GO, PVB-GW-AA-01 (blue spectrum) appears to contain a mixture of plasticizers, as indicated by the additional signals at 173.1, 34.3, 29.2 and 25.3 ppm. Further comparisons with PVBR-E and PVBR-F indicate that this mixture of plasticizers does not include either DHA or DBS. 

Notably, a comparison between the spectra of PVB-GW-NA-04 and PVB-GR-AUTOMOTIVE (Figure 7) shows how, although the rigid part of the materials coincides, the former displays some broadening of the signals of the plasticizer (3GO) with respect to the latter. This is possibly due to a reduction in the mobility of the plasticizer associated with some degree of amorphization [30].

### 3.2. Plasticizer Content by TGA

The satisfactory calculation of the plasticizer content was achieved through the application of TGA (Figure 8). In the case of PVB APS, a single-step curve was obtained, with the onset of degradation determined at 329 °C (*T*_5%_), while the maximum weight loss rate (*T*_d_) was recorded at ca. 391 °C (Table 2). This outcome was expected since the grade is unplasticized; thus, the *T*_5%_ and *T*_d_ correspond exclusively to the decomposition of the polymer. Finally, a char residue of ca. 1 wt.% remained at 600 °C. Similarly, in the case of the pure 3GO (Figure 8a), again, one single step was observed, corresponding to the evaporation of the plasticizer, which began at ca. 188 °C, reached its maximum mass loss rate at ca. 273 °C, and was almost completely evaporated before 300 °C, leaving a low residue of ca. 1.6 wt.%. On the contrary, when looking at the mass loss curves of the plasticized PVB films (PVBR-B, E, and F) a multi-step mass loss curve is observed. A first small step can be found at ca. 100 °C with a mass loss of about 0.5 wt.%, which corresponds to moisture evaporation. The next mass loss step, usually assigned as the first step (Step 1), starts at ca. 150 °C and ends at 330 °C with a mass loss in the range of 18.5–31 wt.%. This step is attributed to the evaporation of the contained plasticizer [23,31], providing an accurate determination of the plasticizer content feasible via TGA. The last mass loss step (Step 2), with a mass loss in the range of 50–75 wt.% takes place between 320 °C and 420 °C and relates to the decomposition of the polymer backbone [26,31].

It is worth mentioning that the plasticizer contained in these samples begins to evaporate in the range of 215–240 °C (*T*_5%_), the latter being ca. 28–55 °C higher than the *T*_5%_ value of the pure 3GO (Table 2). This proves that the plasticizer is effectively embedded in the PVB matrix; thus, its evaporation shifts to higher temperatures. Moreover, the *T*_5%_ values of the PVBR series of samples do not correspond to the onset of polymer degradation, as in the case of unplasticized PVB, but to the onset of plasticizer evaporation. Similarly, the plasticizer removal reaches its maximum rate in the range of 244–310 °C (*T*_d1_), which is higher than the *T*_d_ value of the neat 3GO, but much lower than the neat PVB. The variation in the decomposition temperatures of each PVBR- sample proves that a different plasticizer might affect differently the polymer matrix. More specifically, according to the *T*_d1_ values of PVBR-B, E, and F, it can be concluded that 3GO is the plasticizer more effectively held in the polymer lattice, showing the highest value herein (Table 2). This can be attributed to the more polar character of 3GO, interacting more with the polar groups of the PVB molecular structure. On the contrary, DHA, a molecule of reduced polarity compared to 3GO, is loosely attached to PVBR-E, since it begins to evaporate ca. 74 °C lower. PVBR-F with DBS as plasticizer shows an intermediate behavior closer to that of DHA. In addition, from the deconvolution of the second mass loss step and after subtracting the plasticizer content, the PVB content can be determined accordingly (Table 2). The maximum decomposition rate at the second step (Figure 8) was reached in the range of 382–392 °C (*T*_d2_), very close to the respective value of the unplasticized PVB grade (391 °C). This confirms that the *T*_d2_ values indeed correspond to the decomposition of the PVB fraction. 

Turning to the TGA of GW samples (Figure 8b), it can be deduced that they present the same thermal behavior as the reference samples, with their plasticizer content ranging from 28 to 41% *w*/*w*, depending on the application (automotive or construction) (Table 2).

Moreover, for the analysis of TGA, it is more straightforward to distinguish the two discrete steps of the TGA curve when referring to the derivative (%/°C) of the curves. As shown on the inner graph of Figure 8a, there are two separate peaks, each showing a different decomposition process occurring. Although it is helpful to discriminate the two different processes in progress, the corresponding derivative signals are almost overlapped, making it hard to accurately estimate the limits of each process. To overcome this hurdle and more accurately estimate the plasticizer content, isothermal TGA runs were conducted so as to fully remove the plasticizer from the polymer matrix at a temperature, i.e., 250 °C, where the plasticizer can evaporate, but the PVB fraction would show only very limited mass loss due to thermal degradation (Figure 9a). 

After approximately 70 min at 250 °C, the rate of loss drops markedly, and it might be assumed that complete depletion of the plasticizer has taken place. A continuous slow further weight loss should be attributed to the slow degradation (<4%) of the polymer, as this drop is also observed in the case of the unplasticized polymer. The weight loss due to polymer degradation must be subtracted from the curves of the respective plasticized samples since it is not relative to plasticizer evaporation. In Table 3, the determined plasticizer contents from dynamic and isothermal TGA runs are presented, showing some deviations, especially for PVBR-E. While dynamic analysis provides a better overall understanding of the samples, the isothermal analysis is considered more accurate for the estimation of plasticizer content, but it is more time-consuming and it does not provide further information for the PVB component of the sample (*T*_d2_). Furthermore, the PVBR- series of samples, after the isothermal TGA runs, were scanned dynamically up to 600 °C (Figure 9b). Indeed, the mass-loss curves of all PVBR- became one-step, and mass loss in the range of 200–300 °C is no longer observed, thus proving that the plasticizer was efficiently and totally removed during the previous isothermal step.

In order to verify the aforementioned TGA results on the plasticizer content, direct removal of the plasticizer was attempted by SE, which is a well-known process to extract oils or substances from PVB samples in many cases [22,24]. The process was applied on all the reference plasticized PVB samples (PVBR-B, E, F) for removing the plasticizers (3GO, DHA, DBS, respectively), and determining the mass balance. Plasticizers can be either internal or external. 3GO, DHA, and DBS are considered mainly as external plasticizers since they can be lost by evaporation, migration, or extraction [31]. Hexane was used as a solvent, due to its relatively low boiling point (ca. 69 °C) and its miscibility with the contained plasticizers [25]. All three PVBR- samples were subjected to SE for 12 h, after which the residual PVBR-SE samples were also characterized by TGA analysis. Accordingly (Table 4), PVBR-E-SE and F-SE samples showed plasticizer removal values very close to the pre-determined values by TGA. Indeed, as shown by TGA analysis (Figure 10) in the SE-treated samples, the curves became one step; thus, total removal of DHA or DBS should have been achieved. On the contrary, in the case of PVBR-B-SE, which contains 3GO, total removal was not successful since a two-step decomposition was still observed, with the 1st step determined at ca. 12%, which is much lower than the initial value of 31% *w*/*w*, that PVBR-B exhibited (Table 3). 3GO seems to display a higher affinity for the polymeric matrix than for hexane in comparison with DBS and DHA, since the boiling points of these three plasticizers are very similar (in the range of 344–350 °C). The different polarity of each plasticizer could be an explanation, too, since 3GO has a different and more polar molecular structure as previously explained, compared to DHA and DBS, which are more similar to each other [8].

### 3.3. Plasticizer Content by FT-IR Analysis

Alternatively, for a quicker and nondestructive estimation of the plasticizer content, FT-IR was used. According to the aforementioned FT-IR analysis, the qualitative identification of the main PVB and plasticizer peaks and plasticizer is feasible. In order to have a quantitative view of the plasticizer content, the development of an FT-IR master curve was performed, which was calibrated against the signal of the plasticizer peak at 1740 cm^−1^ [32]. To calibrate the master curve, five model samples of predefined plasticizer content were prepared by casting in THF. Accordingly, unplasticized PVB APS was solubilized in THF, and each time the desired 3GO fraction (0, 10, 20, 30, 40 wt.%) was added to the formed solution. After stirring, each solution was placed in a petri dish and was left overnight under the fume hood so as to evaporate the solvent and produce the casted film. Subsequently, the received films were further vacuum dried in the oven at 50 °C for total removal of THF. The thickness of the casting membranes varied, but this is no issue for ATR, since it estimates the superficial concentration of a solid and since all the films are homogenous. In addition, 10 different FT-IR spectra at different spots were received for each model-casted sample, and the average intensity of the C=O stretching peak at 1740 cm^−1^ was determined. The extracted FT-IR master curve is shown in Figure 11, where there is a clear linear equation (R^2^ = 0.99) *y* = 209.27 × *x*, where *y* is the plasticizer content and *x* is the determined absorbance. The intercept was assumed to be zero, since the VAc content is too low in most PVB to contribute to the estimations for most PVB products.

Assuming that most plasticizers exhibit the same absorbance intensity at 1740 cm^−1^ as 3GO and that most PVB grades have a similar VAc content (<3%), the FT-IR master curve was tested for all the samples, and it was concluded that it could reliably estimate the plasticizer content within a deviation of ±6 wt.% (as reported above in Table 4). This is considered a useful tool for a quick and fair estimation of the plasticizer content of a PVB film. Furthermore, in order tο ensure that membranes contained the actual dosed plasticizer content, determination of the real plasticizer quantity via dynamic TGA was performed, while the *T*_g_ value via DSC was determined and correlated to the plasticizer content. As expected, the 3GO content and determined *T*_g_ value were inversely proportional according to DSC, while the 3GO content as determined by TGA matched the actual dosed quantity (Figure 12). 

### 3.4. Plasticizer Content by NMR

Aiming at a more chemistry-oriented investigation, liquid-phase NMR was exploited (Figure 13), a powerful technique to analyze the chemical composition of a composite material (such as PVB) and/or to quantify the relative amounts of different components in a specific formulation (i.e., the presence of a plasticizer). In the present context, NMR would also be very useful to assess possible degradation suffered by the polymeric film undergoing different treatments. All the values discussed in the following paragraphs are summarized in Table 5.

By comparing the integrated area under the NMR peaks, it is feasible to calculate the relative concentration of two compounds in a sample. All solution NMR spectra of the PVBR-R samples show the presence of the characteristic PVB signal, located at 4.53 ppm, along with the PV-OH (vinyl-alcohol residue) one at 4.28 ppm. The presence of PV-OH is expected as the reaction between polyvinyl alcohol and butyraldehyde is not quantitative. Quite similar PV-OH:PVB: molar (weight) ratios were found, slightly increasing from PVBR-B, 1:7.7 (4:96), to PVBR-F, 1:6.7 (4:96), to PVBR-E, 1:5.9 (5:95). It should be noted that these differences are too small to be considered significant. On the other hand, spectra differ in the peaks ascribable to the different plasticizers. PVBR-B shows a characteristic peak of 3GO located at 4.23 ppm, whose integration allows calculating the polymer-to-plasticizer molar (weight) ratio as high as 7.7 (33:67). Here, it is worth recalling that the weight ratio has been calculated using the molecular weight of the vinyl butyrate (or vinyl alcohol) monomeric unit. Only a slightly higher amount of a different plasticizer was found in PVBR-E (signals at 4.07 and 2.34 ppm for DHA) and PVBR-F (signals at 4.07 and 4.31 ppm for DBS) with a 34:66 and 41:59 weight ratio, respectively. It is worth noting that the molar ratios (i.e., 1:5.3 and 1:3.8, respectively) are more variable, being normalized for the (different) molecular weight of the plasticizers (i.e., 402.6 for 3GO and 314.67 g mol^−1^ for both DBS and DHA). As distinguishing evidence, PVBR-F clearly shows the presence of some spurious peaks (highlighted with yellow circles in Figure 13), ascribable to an additional (very likely 3GO) plasticizer accounting for 8% of the total plasticizer amount. With respect to plasticizer quantification, a remarkably good agreement with TGA data was found for PVBR-B only, whereas the plasticizer amount for both PVBR-E and PVBR-F resulted quite overestimated (Table 3). Such a discrepancy could be justified considering different factors both from the TGA and NMR points of view. Starting from the latter, the overestimation found in the analyses could be related (i) to different relaxation times between the polymer and the plasticizer, showing much longer values or (ii) to a different solubility of the two components in the NMR solvent (MeOD—Deuterated methanol). Thus, we performed some experiments increasing the relaxation time (up to 60 s) or reducing the concentration (down to 5 mg/mL); yet, in both cases, no significant modification of the NMR spectra could be evidenced. On the other hand, an underestimation of plasticizer amount from TGA data could be ascribable to a sequestration of the plasticizer into the polymer structure: as a matter of fact, “entrapped” plasticizer will evaporate only when the associated structure of the polymer is broken. 

Finally, some peaks in the aromatic region (between 6 and 9 ppm) are detected for all the analyzed spectra but with a remarkably weaker intensity compared to the main peaks (<0.1%). They were tentatively assigned to thermal and/or UV stabilizers (usually heterocyclic compounds) that could be likely present within the polymer formulation. 

When glass waste samples were analyzed (Figure 14), only minor changes could be observed, at least from a qualitative point of view, in the NMR spectra but for PVB-GW-AA-01. Indeed, in its spectrum 3GO appears as the secondary plasticizer, with the main one being different, generating characteristic peaks located at 4.21 ppm (multiplet) and 2.33 ppm (triplet). This is also associated with qualitative differences in the multiplet accounting for the aliphatic protons resonating below 1 ppm and around 1.2–1.3 ppm. Albeit it was not possible to figure out the nature of these plasticizers, due to the similarity in both chemical shift and shape of its signals compared to the homologous 3GO, we can propose two possible attributions: it could be a structural analogue of 3GO or a degradation product of the latter. On the other hand, PVB-GR-AUTOMOTIVE only presents a spurious peak (barely visible) at 3.49 ppm (quartet), whereas no meaningful difference in PVB-GW-C-01 and PVB-GW-NA-04 spectra, compared to the pristine PVB material, could be observed.

Going into detail with the quantification of both PVB and plasticizers, one could note that the relative amount of the latter increases, passing from PVB-GW-AA-01 (PVB:PL weight ratio = 1:0.22), to PVB-GW-C-01 (1:0.29), to PVB-GW-NA-04 (1:0.34), and reaching a maximum plasticizer amount in PVB-GR-AUTOMOTIVE (1:0.34). A similar trend could also be seen when PV-OH is considered (Table 5).

### 3.5. Molecular Weight 

Dilute solution viscosity analysis is an accurate analytical tool for indirectly estimating the molecular weight of polymers. Solution viscometry was performed in the case of PVB films in tetrahydrofuran (THF) [20]. The concentration of the solutions is 0.5 g·dL^−1^ [33], but in the case of measuring plasticized PVB grades, the concentration of the solution was corrected for the plasticizer content as determined by TGA analysis, so as to have an actual polymer concentration of 0.5 wt.%. That way, any impact of the plasticizer on the viscosity was reduced. In fact, this viscosity statement is only true if there is no close interaction between the polymer and the plasticizer in the diluted solution. If the hydrodynamic volume is changed through interactions the determined value is not any more correct. Nevertheless, the particular analytical technique is considered reliable to relatively compare the molecular weight of the different PVB samples. PVB APS exhibited an [*η*] value of 1.75 dL g^−1^, while the plasticized grades were determined in the range of 1.50–2.02 dL g^−1^ (Table 6). 

To clarify this, supplementary GPC measurements in THF were also performed (Figure 15). As shown, there are two different distributions; the first corresponds to the higher molecular weight of polymer (PVB), and the second one to the plasticizer moiety. On the contrary, for the unplasticized PVB samples (PVB APS) only one peak corresponding to the polymer was determined. The viscosity average molecular weights (*Μ_ν_*) of the tested samples were found in the range of 95,000–135,000 g mol^−1^ and correlates nicely to the trend of intrinsic viscosity as determined by DSV (Table 6) only for the case of the neat PVB grades.

Apart from qualitative results for the molecular weight of PVB, it is worthwhile mentioning that the chromatographs of plasticized PVB samples can be exploited for the identification of the plasticizer type. As shown in Figure 15, the retention time for the second region differs in each sample, pinpointing the different plasticizers used. 3GO (402.6 g/mol), is a moiety with a higher molecular weight than DHA (314.5) g/mol) and DBS (314.4 g/mol), since it eludes earlier than the other two. In this way, GPC measurements again verify the presence of 3GO in PVBR-B, found by NMR, and show that it is also feasible to identify the plasticizer type, by comparing the retention times for the second part of the chromatograph. Furthermore, a fair estimation of the plasticizer content that correlates quite well to the isothermal TGA values (Table 3) can be made from the plasticizer peaks at the GPC graph, either by the intensity of the peaks or by the area under the peaks, corrected to each individual response. 

### 3.6. Rheology

The MFR was measured for PVB APS at 190 °C with a weight of 10 kg, and the reference PVB films were measured at 190 °C and 2.16 kg (Table 7). PVB APS exhibited the lowest melt flow rate (2.16 kg weight was not sufficient to induce flow; therefore, 10 kg was used), since it contains no plasticizer. Regarding plasticized PVB samples (reference and waste), in most cases, the higher the plasticizer content the higher the MFR, which is expected. Nonetheless, this is not always the rule since PVBR-E and F, although their noteworthy plasticizer content (15.6% and 19.7%, respectively), exhibited relatively low MFRs (0.71 and 0.19 g/10 min), proving that the plasticizer is not the only parameter affecting the MFR. This fact could indirectly indicate that the initial PVB grade used might have been of a higher molecular weight. Indeed, as determined by solution viscometry and gel permeation chromatography in THF, PVBR-E, and F exhibited a higher MW than PVBR-B. Apart from that, it should also be stated that the difference in plasticizers’ nature (3GO, DBS, and DHA) could also affect the MFR [34,35]. The latter also stands for the waste samples PVB-GW-NA-04 and PVB-GW-AA 01. Based on this, no safe conclusion regarding the effect of aging on the MFR can be made, even if the aged automotive grade (GW-AA-01) shows a higher value compared to the nonaged grade (GW-NA-04).

## 4. Conclusions

In this work, a characterization strategy for PVB was developed aiming at benchmarking the key properties of raw grades appropriate for interlayer use, which will be further exploited for the sorting of post-consumed PVB grades. Accordingly, analytical techniques, such as TGA, FT-IR, and NMR, were applied so as to study plasticizer type and content, molecular size, and rheological behavior, and vinyl butyral and vinyl alcohol fractions. TGA provides an accurate estimation of the plasticizer content, but cannot provide needed insights as regards the type of plasticizer. In dynamic TGA measurements, the limits between plasticizer evaporation and PVB decomposition are not always clear. Alternatively, isothermal TGA measurements at an appropriate temperature that allows for plasticizer evaporation while PVB decomposition is largely avoided are considered more reliable for the accurate quantification of the plasticizer content. Upon development of a FT-IR master curve, the determination of plasticizer content is also possible; compared to the TGA-based method, the one in consideration is somewhat less accurate but faster and nondestructive, while we obtain some information about the plasticizer composition on the basis of identifying f certain chemical groups (e.g., it can be concluded that the plasticizer is an oligo-ester). NMR offers information sufficient for the identification of common plasticizers while it leads to overestimation of the plasticizer content in comparison to TGA. GPC provides accurate information of the molecular weight of PVB, but also can give qualitative information about the plasticizer content. Finally, MFR values correlate quite well to the determined plasticizer content but cannot give additional info regarding potential aging suffered in the waste samples. Overall, the present work suggests a good practice for the in-depth analysis of the interlayers of the laminated glass, and proposes an industrial-friendly methodology for the off-line classification of the type of interlayers in waste laminated glass for wise and sustainable recycling. Critical data are herein obtained that can be further exploited for the development of an on-line sorting sensor of PVB wastes.

## Figures and Tables

**Figure 1 polymers-16-00010-f001:**
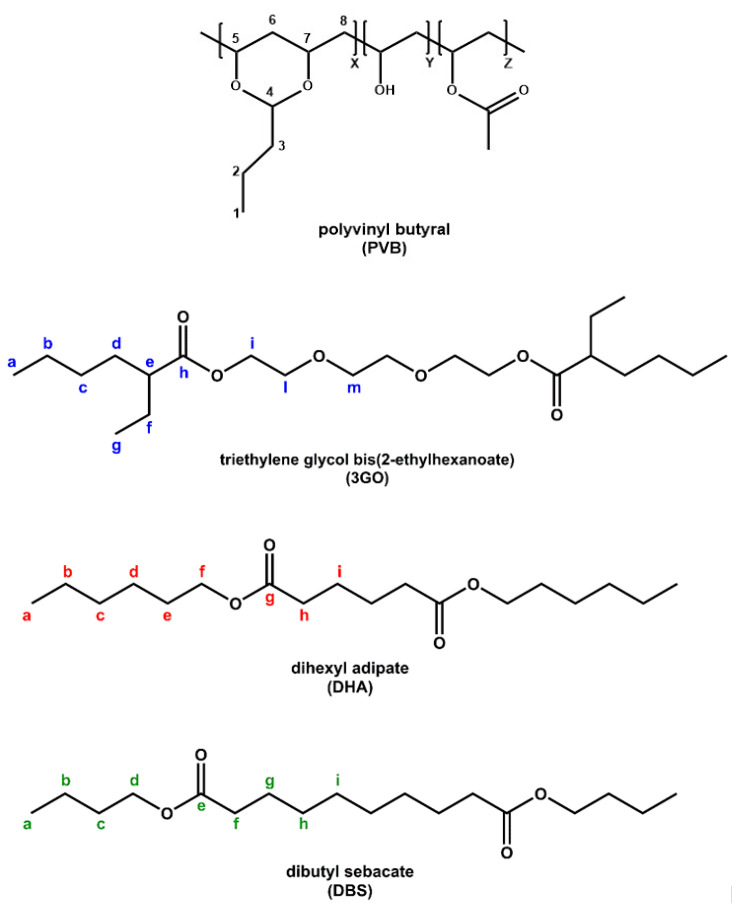
From the top, first: terpolymer molecular structure of PVB (X = VB, Y = VA, Z = VAc); second: molecular structure of 3GO; third: molecular structure of DHA; fourth: molecular structure of DBS. The main C atoms for all compounds are numbered/labelled. Different colours are used to distinguish labelling for the different compounds.

**Figure 2 polymers-16-00010-f002:**
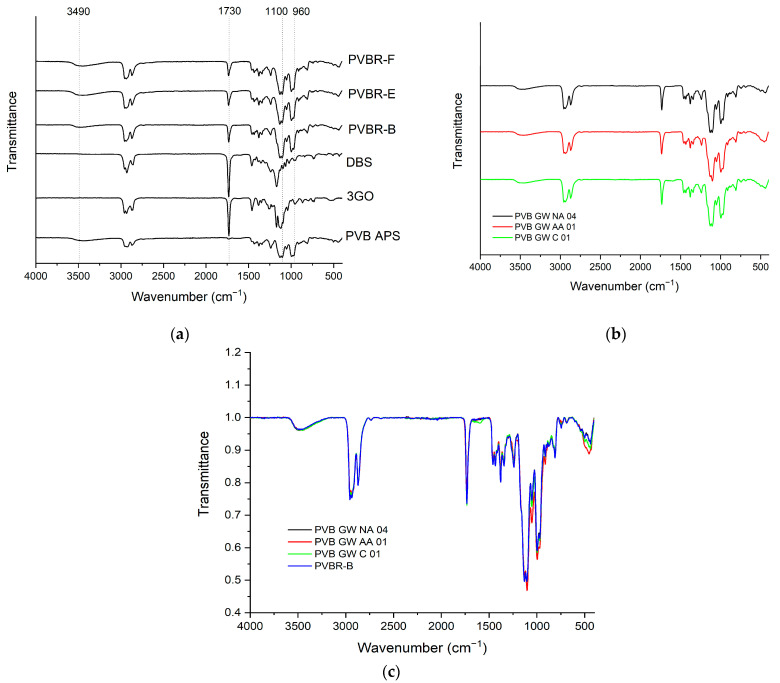
(**a**) Comparative FT-IR spectra of neat reference PVB grades and plasticizers; (**b**) FTIR Spectra of PVB waste grades; (**c**) Comparison between neat reference PVBR-B and PVB waste grades.

**Figure 3 polymers-16-00010-f003:**
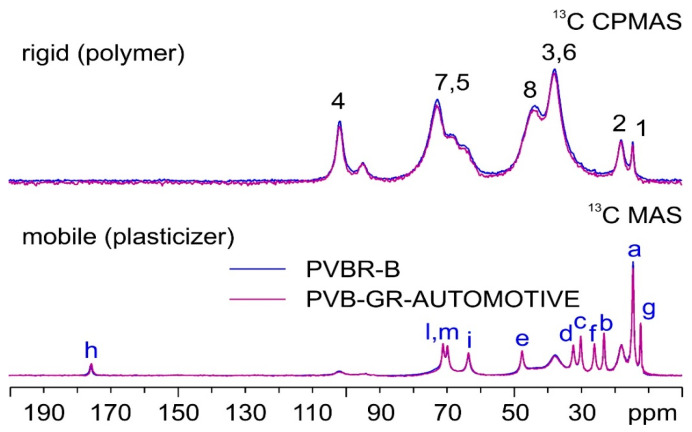
^13^C (100.63 MHz) CPMAS (**top**) and MAS (**bottom**) spectra of PVBR-B (in blue) and PVB-GR-AUTOMOTIVE (in purple), acquired at a spinning speed of 8 kHz at room temperature. Letters above the signals correspond to the peak assignments (please refer to Figure 1 for atom numbering/labeling).

**Figure 4 polymers-16-00010-f004:**
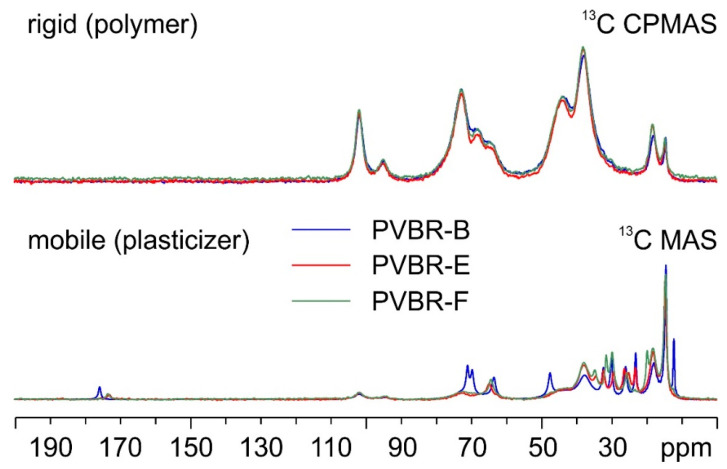
^13^C (100.63 MHz) CPMAS (**top**) and MAS (**bottom**) spectra of PVBR-B (in blue), PVBR-E (in red), and PVBR-F (in green), acquired at a spinning speed of 8 kHz at room temperature.

**Figure 5 polymers-16-00010-f005:**
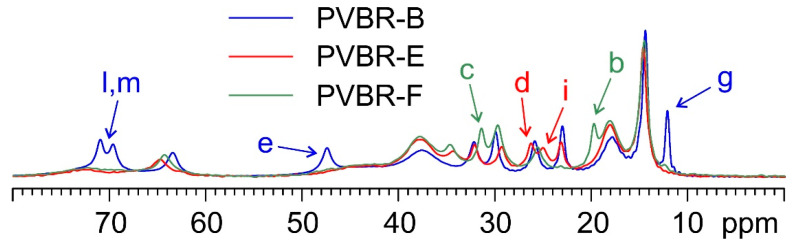
Detail of the ^13^C MAS spectra of PVBR-B, PVBR-E, and PVBR-F. Colored arrows (blue: 3GO; red: DHA; green: DBS) indicate assigned diagnostic signals, specific to the plasticizer contained in the sample (please refer to Figure 1 for atom labeling).

**Figure 6 polymers-16-00010-f006:**
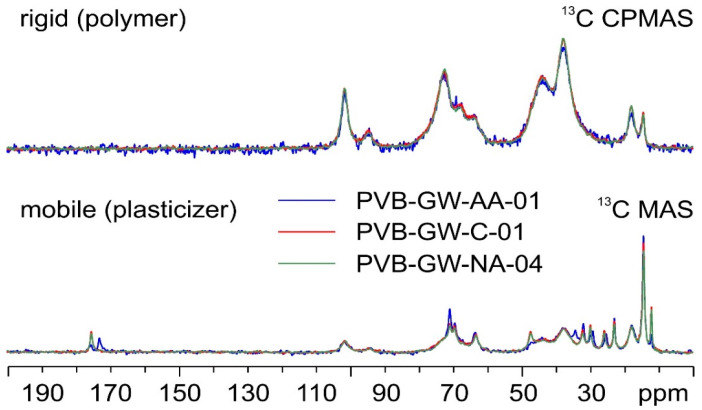
^13^C (100.63 MHz) CPMAS (**top**) and MAS (**bottom**) spectra of PVB-GW-AA-01 (in blue), PVB-GW-C-01 (in red), and PVB-GW-NA-04 (in green), acquired at a spinning speed of 8 kHz at room temperature.

**Figure 7 polymers-16-00010-f007:**
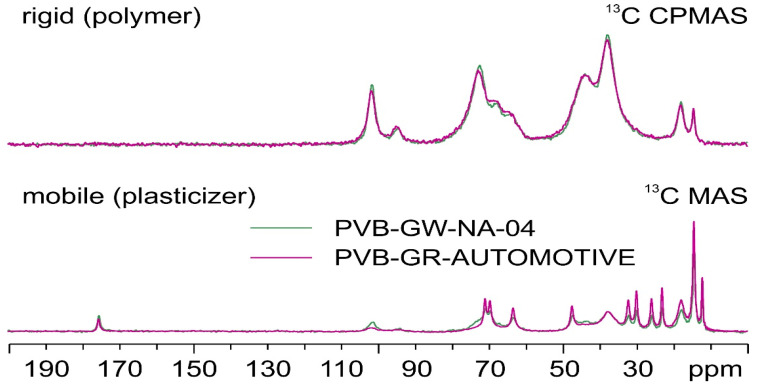
^13^C (100.63 MHz) CPMAS (**top**) and DE MAS (**bottom**) spectra of PVB-GW-NA-04 (in green) and PVB-GR-AUTOMOTIVE (in purple), acquired at a spinning speed of 8 kHz at room temperature.

**Figure 8 polymers-16-00010-f008:**
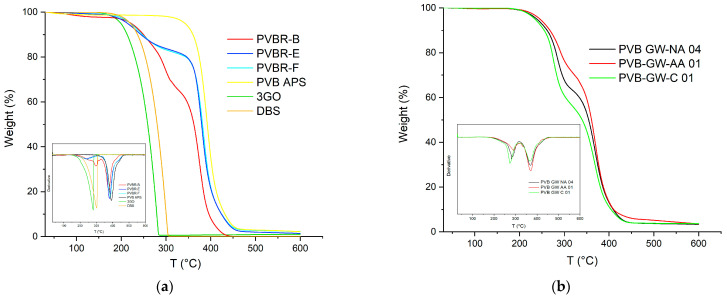
(**a**) Mass loss curves of PVB grades; (**b**) and derivative TGA curves.

**Figure 9 polymers-16-00010-f009:**
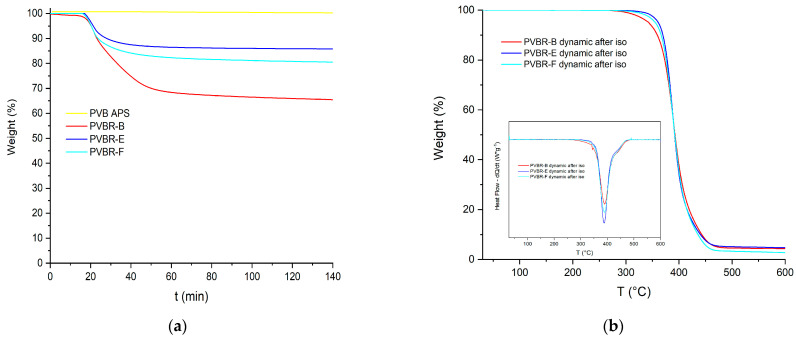
(**a**) Isothermal TGA runs at 250 °C; (**b**) dynamic TGA runs on the residue of isothermal runs.

**Figure 10 polymers-16-00010-f010:**
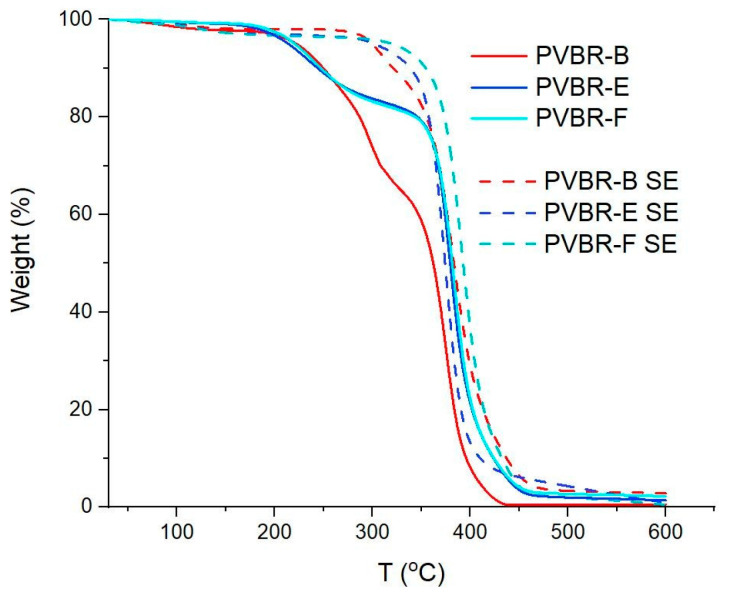
Comparative TGA curves for samples before and after SE.

**Figure 11 polymers-16-00010-f011:**
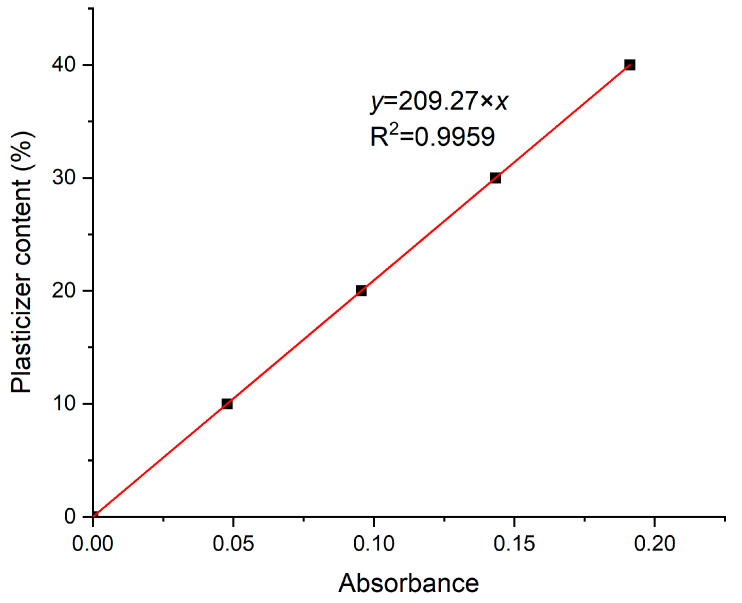
Development of FT-IR master curve calibrated against the intensity of the carbonyl peak at 1740 cm^−1^.

**Figure 12 polymers-16-00010-f012:**
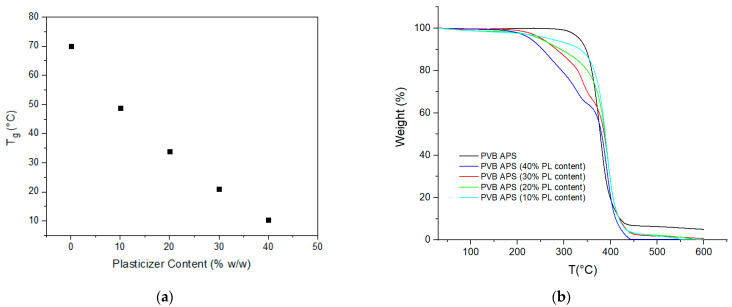
(**a**) Model-casted PVB films were used for the calibration of the FT-IR master curve. T_g_ values of the model-casted PVB films as a function of plasticizer content; (**b**) dynamic TGA curves of the model-casted PVB films (right).

**Figure 13 polymers-16-00010-f013:**
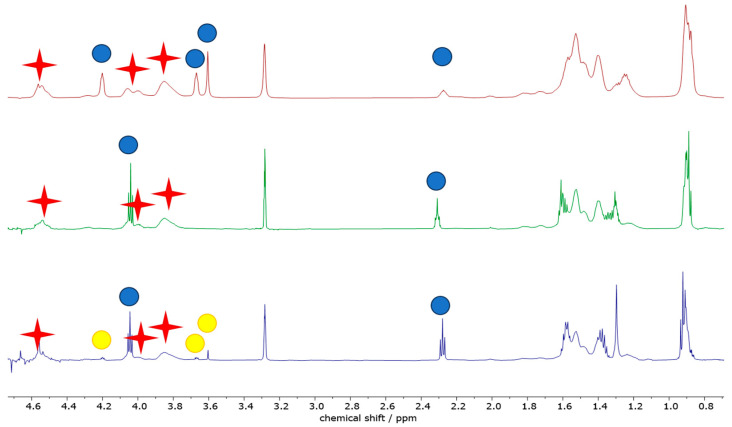
^1^H (600.91 MHz) solution (CD_3_OD) NMR spectra (0.6–4.7 ppm region) of PVBR-B (**top**, in red), PVBR-E (**middle**, in green) and PVBR-F (**bottom**, in blue). Red crosses and blue circles highlight the characteristic peaks of the polymer and the plasticizers, respectively. Yellow circles highlight the characteristic peaks of the spurious plasticizers.

**Figure 14 polymers-16-00010-f014:**
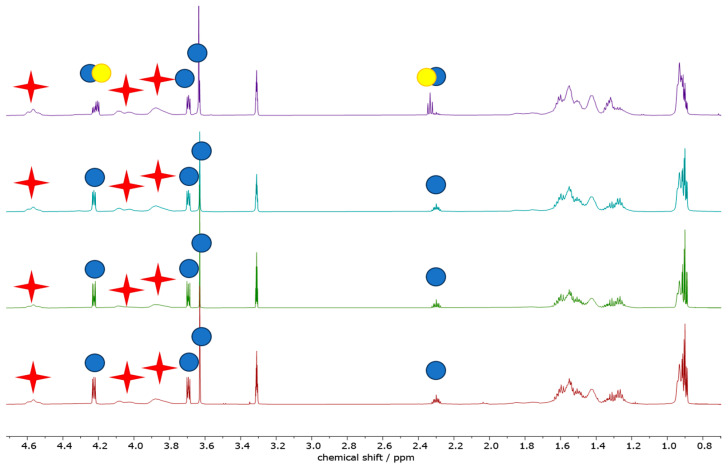
Zoomed (0.6–4.8 ppm) ^1^H (600.91 MHz) solution (CD_3_OD) NMR spectra of PVB-GW-AA-01 (**top**, in purple), PVB-GW-C-01 (**middle top**, in turquoise), PVB-GW-NA-04 (**middle bottom**, in green) and PVB-GR-AUTOMOTIVE (**bottom**, in red). Red crosses and blue circles highlight the characteristic peaks of the polymer and the plasticizers, respectively. Yellow circles highlight the characteristic peaks of the unknown plasticizer.

**Figure 15 polymers-16-00010-f015:**
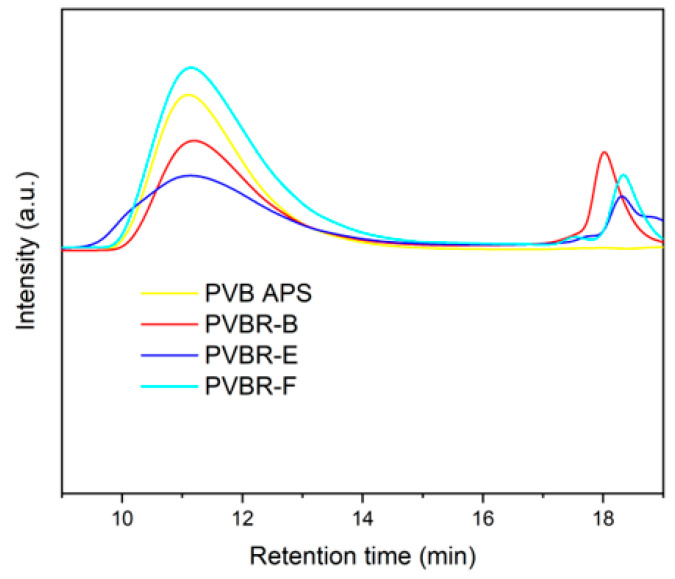
GPC chromatograph of PVB reference samples.

**Table 1 polymers-16-00010-t001:** Description of the samples used in the present study.

Sample	Plasticizer (Short Name)	Manufacturer	Commercial Name
PVB APS	Unplasticized	American Polymer Standards Corporation, Mentor, OH, USA	-
PVBR-B	Triethylene glycol bis(2-ethylhexanoate) (3GO)	Eastman, Kingsport, Tennessee, Kingsport, TN, USA	Saflex RB41
PVBR-E	Dihexyl adipate (DHA)		Saflex AG
PVBR-F	Dibutyl sebacate (DBS)		Saflex DB
PVB-GW-NA 04	to be determined	Waste from automotive	-
PVB-GW-AA 01			-
PVB-GW-C 01		Waste from construction	-
PVB-GR-AUTOMOTIVE		PVB extracted from a new windshield	-

**Table 2 polymers-16-00010-t002:** Data from dynamic TGA measurements.

Samples	*T*_5%_ (°C)	*T*_d1_ (°C)	Step 1 (% *w*/*w*)	*T*_d2_ (°C)	Step 2 (% *w*/*w*)	Residue (% *w*/*w*)
		Plasticizer Evaporation		PVB Decomposition		
PVB APS	329.1 ± 13.1	-	-	391.1 ± 0.4	99.0 ± 1.0	1.0 ± 1.0
3GO	187.5 ± 6.2	272.9 ± 24.0	98.4 ± 1.2	-	-	1.6 ± 1.2
DBS	213.7 ± 6.7	219.9 ± 7.6	99.6 ± 0.1	-	-	0.3 ± 0.1
PVBR-B	215.6 ± 4.1	310.2 ± 0.9	30.7 ± 1.0	392.0 ± 0.3	64.0 ± 1.8	1.4 ± 1.4
PVBR-E	204.1 ± 1.6	236.2 ± 0.6	21.7 ± 2.7	370.7 ± 9.2	76.9 ± 0.1	0.7 ± 0.7
PVBR-F	225.1 ± 3.3	244.3 ± 2.3	18.5 ± 0.6	382.4 ± 1.6	77.5 ± 0.8	2.4 ± 0.2
PVB-GW-NA 04	235.0 ± 3.8	295.4 ± 1.6	36.6 ± 0.1	386.9 ± 1.9	57.5 ± 0.1	3.8 ± 0.4
PVB-GW-AA 01	245.8 ± 0.5	306.9 ± 3.7	28.1 ± 1.1	393.3 ± 5.0	65.2 ± 1.4	5.3 ± 0.1
PVB-GW-C 01	237.5 ± 2.0	289.3 ± 0.5	40.8 ± 2.5	387.2 ± 0.4	53.7 ± 1.4	4.6 ± 1.0

**Table 3 polymers-16-00010-t003:** Plasticizer content as determined by dynamic and isothermal TGA runs.

Samples	Plasticizer Content (%) TGA Dynamic Tests	Plasticizer Content (%) TGA Isothermal Tests	Plasticizer Content (%) FT-IR Mastercurve
PVBR-B	30.7 ± 1.0	31.0 ± 0.9	28.1
PVBR-E	21.7 ± 2.7	15.6 ± 0.1	20.4
PVBR-F	18.5 ± 0.6	19.7 ± 0.5	21.3

**Table 4 polymers-16-00010-t004:** Plasticizer content determined after SE.

Samples	Plasticizer Lost after SE (%)	Plasticizer Content by Dynamic Analysis TGA (%)
PVBR-B SE	21.3	12.0
PVBR-E SE	18.1	0
PVBR-F SE	17.7	0

**Table 5 polymers-16-00010-t005:** Quantitative analyses of NMR spectra normalized by setting the integral of PVB at 4.53 ppm equal to 1. The integral of a signal at 4.28 ppm has been used to quantify the amount of PVB. The chemical shifts used for the quantification of the plasticizers are cited in the text.

Sample	Integral Plasticizer	Molar Ratio PVB/Plasticizer	Integral PVB-OH	Molar Ratio PVB/PVB-OH	Weight Ratio PVB/Plasticizer (PVB:Plasticizer)	Weight Ratio (PVB + PVOH)/Plasticizer (PVB + PVOH:PLasticizer)
PVBR-B	0.18	5.6	0.13	7.7	2.00 (67:33)	2.07 (67:33)
PVBR-E	0.19	5.3	0.17	5.9	1.86 (65:35)	1.96 (66:34)
PVBR-F	0.21 + 0.05	3.8	0.15	6.7	1.38 (58:42)	1.42 (59:41)
PVB-GW-NA-04	0.32	3.1	0.14	7.1	1.11 (53:47)	1.17 (54:46)
PVB-GW-AA-01	0.22 *	4.5	0.18	5.6	1.62 (62:38)	1.72 (63:37)
PVB-GW-C-01	0.29	3.4	0.20	5.0	1.23 (55:45)	1.31 (57:43)
PVB-GR-AUTOMOTIVE	0.34	2.9	0.18	5.6	1.02 (50:50)	1.08 (52:48)

* This value refers to the sum of both detected plasticizers.

**Table 6 polymers-16-00010-t006:** Determined intrinsic viscosity values of herein characterized PVB resins.

Samples	Plasticizer Content (%)	[*η*] (dL/g) by DSV	${\bar{M}}_{v}$ (g/mol) by GPC	PDI for PVB
PVB APS	0	1.75 ± 0.01	122,000	1.76
PVBR-B	31	1.51 ± 0.09	110,000	1.81
PVBR-E	22	1.90 ± 0.01	132,040	2.06
PVBR-F	19	1.86 ± 0.11	114,000	2.03
PVB GW NA 04	37	1.89	105,000	1.83
PVB GW AA 01	28	1.50	99,000	1.77
PVB GW C 01	41	2.02	107,000	1.87

**Table 7 polymers-16-00010-t007:** Melt Flow Rates (MFRs) of herein characterized PVB resins.

Samples	PL Content (%)	MFR (g/10 min)
PVB APS *	0	0.40
PVBR-B **	31	1.69
PVBR-E **	15.6	0.71
PVBR-F **	19.7	0.19
PVB-GW-NA-04 **	37	1.72
PVB-GW-AA 01 **	28	2.25
PVB-GW-C 01 **	41	2.45

* measured with 10 kg weight, ** measured with 2.16 kg weight.

## Data Availability

Data is contained within the article.

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
