# Peer review of "An Integrated Characterization Strategy on Board for Recycling of poly(vinyl butyral) (PVB) from Laminated Glass Wastes"

_polymers, 2023, doi:10.3390/polym16010010_

Round 1
Reviewer 1 Report
Comments and Suggestions for Authors
In the paper, the authors made various chemical analyzes for PVB, which contains various plasticizers, and extracted PVB from waste in the automotive industry and construction, as well as from windshields. The work is well structured and analyzes were made of which plastifiers are present and what their content is with various methods. I have to admit that from the title I expected more to be written about recycling itself, because that's what the title suggested. I suggest that the title be reformulated because this way for recycling of PVB from laminated glass wastes suggests that the process for recycling and separating PVB from other glass will also be described in the paper.
The passage from Line 89 to 97 should also be rewritten. This does not give the impression of what you have done in science. And that it is then connected with the conclusion. Definitely reinforce the explanation to get readers interested.
Author Response
Reviewer 1
In the paper, the authors made various chemical analyzes for PVB, which contains various plasticizers, and extracted PVB from waste in the automotive industry and construction, as well as from windshields. The work is well structured and analyzes were made of which plastifiers are present and what their content is with various methods. I have to admit that from the title I expected more to be written about recycling itself, because that's what the title suggested. I suggest that the title be reformulated because this way for recycling of PVB from laminated glass wastes suggests that the process for recycling and separating PVB from other glass will also be described in the paper.
The passage from Line 89 to 97 should also be rewritten. This does not give the impression of what you have done in science. And that it is then connected with the conclusion. Definitely reinforce the explanation to get readers interested.
Point 1: I suggest that the title be reformulated because this way for recycling of PVB from laminated glass wastes suggests that the process for recycling and separating PVB from other glass will also be described in the paper.
Response:
The authors suggest the following alternative title for the paper: ‘An integrated characterization strategy on board for recycling of PVB from laminated glass wastes’.
The title was modified in order to remain in consonance with SUNRISE H2020 project (grant agreement No 958243). It is essential to emphasize that this work is a benchmarking strategy for the determination of critical properties of PVB intended for recycling. That is why a comparison of critical properties of commercial neat reference PVB grades and PVB recyclates derived from automotive or construction is described in the paper, in order to determine the prerequisite parameters that the recyclate should meet so as to be re-used for the same interlayer application. Therefore, this publication is considered as the first approach and as an indivisible part of a broader recycling plan of PVB from laminated glass wastes, which will produce later more publications οn this field.
Point 2:
The passage from Line 89 to 97 should also be rewritten. This does not give the impression of what you have done in science. And that it is then connected with the conclusion. Definitely reinforce the explanation to get readers interested.
Response:
It was attempted to create a more pronounced significance to the aim of this work. The referred passage has been modified and highlighted in yellow in the text. More specifically the updated text is the following:
‘…..Up to now, most of the post-consume PVB interlayer material is incinerated or landfilled, and only a 9 % is recycled in secondary uses….’
‘..Retrieved PVB should be also sorted into categories (e.g., plasticizer type and/or content, extent of degradation) for its better management and optimum recycling. In this concept, an innovation of the present work relies on an integrated characterization strategy proposed as a preliminary step to the recycling of PVB. Identifying the different grades of PVB and the most common used plasticizers will enable the sorting of laminated glass for PVB recycling. Novel strategies based on the combination of several advanced analytical techniques and the application of spectral data are proposed to provide a more complete evaluation about the composition and degradation level of the laminated layer.
Based on this holistic approach, in the current study, a part of a strategy for the high-quality recycling and valorization of PVB from laminated glass into new interlayer material is demonstrated. We aim to set the basis for the categorization of various PVB grades and films by characterizing different commercial PVB reference material through application of several methods, namely FT-IR (Fourier-transform Infrared spectroscopy), solution and solid-state NMR (Nuclear Magnetic Resonance), TGA (Thermogravimetric Analysis), DSC (Differential Scanning Calorimetry), MFR (Melt Flow Rate) and DSV (dilute solution viscometry). To the best of our knowledge, the combination of different techniques has not been applied, for the development of an accurate in-line, rapid and robust inspection solution for the characterization of the composition and quality of PVB films in laminated glass. Benchmarking of the critical properties of reference PVB samples will allow the determination of the optimum level of properties that a post-consumed grade should meet, in order to be re-used as an interlayer material (“closed loop”).’

Reviewer 2 Report
Comments and Suggestions for Authors
The study presented in this manuscript proposes an exhaustive characterization strategy based on a combination of analytical techniques such as TGA, FT-IR and NMR to characterize prime and recycled PVB from laminated glass wastes in order to determine plasticizer type and content, molecular weight and rheological behavior, that can be exploited for the sorting of post-consumed PVB grades. The topic studied is very relevant to current challenges of sorting and recycling. I recommend the publication of this manuscript in Polymers after addressing the following minor comments:
- It would be great to combine FTIr results in Figure 2a and Figure 2b to ilustrate that there is small/minor variation between reference and recycled PVB grades in terms of FTIR spectra.
- I recommend measuring the MFR of PVB APS at the same conditions.
Comments on the Quality of English Language
Minor editing of English language is recommended
Author Response
Reviewer 2
The study presented in this manuscript proposes an exhaustive characterization strategy based on a combination of analytical techniques such as TGA, FT-IR and NMR to characterize prime and recycled PVB from laminated glass wastes in order to determine plasticizer type and content, molecular weight and rheological behavior, that can be exploited for the sorting of post-consumed PVB grades. The topic studied is very relevant to current challenges of sorting and recycling. I recommend the publication of this manuscript in Polymers after addressing the following minor comments:
- It would be great to combine FTIr results in Figure 2a and Figure 2b to ilustrate that there is small/minor variation between reference and recycled PVB grades in terms of FTIR spectra.
- I recommend measuring the MFR of PVB APS at the same conditions.
Point 1: It would be great to combine FTIr results in Figure 2a and Figure 2b to ilustrate that there is small/minor variation between reference and recycled PVB grades in terms of FTIR spectra.
Response: A comparison was added as a Figure 2c and a corresponding comment was added to the manuscript.
‘Regarding the FT-IR spectra analysis of GW samples (Figure 2 b), there is no clear difference from the reference samples (PVBR). To clarify that, PVB waste grades were compared to PVBR-B (Figure 2 c), which all of them contain 3GO as plasticizer (as shown by NMR analysis below). An overlapping of the characteristic peaks and minor differentiation in the intensity of the peaks were only observed, proving there is no great chemical variation between PVB films after their end of life. Therefore FT-IR cannot be used for the study of ageing or degradation phenomena of this type and extent that are typical for the waste materials of practical interest.’
Point 2 : I recommend measuring the MFR of PVB APS at the same conditions.
Response: PVB APS is unplasticized and it exhibits no melt flow with the same conditions (2.16 kg). The weight of 2.16 kg is not sufficient to induce a flow to the melt polymer. That is why we switched to a 10 kg load, so as to observed flow. A relevant explanation was added to the text in order to help the better understanding of the reader:
‘…2.16 kg weight was not sufficient to induce flow; therefore 10 kg was used ...’

Reviewer 3 Report
Comments and Suggestions for Authors
In the manuscript entitled "An integrated characterization strategy for recycling of PVB from laminated glass wastes" the authors have conducted a thorough characterization of PVB films. They reported a method for recycling and reuse of post-consumed PVB grades from automotive and construction sector. The plasticizer content and their chemical nature also determined authors.
The manuscript is well written and well supported by the data. Only few minor things should be corrected before the acceptance of the paper.
For example, Eq1 should be in high resolution.
The inset images of Figure 8 is not clear, please make them more big and high resolution. Same for Figure 9b also.
Author Response
Reviewer 3
In the manuscript entitled "An integrated characterization strategy for recycling of PVB from laminated glass wastes" the authors have conducted a thorough characterization of PVB films. They reported a method for recycling and reuse of post-consumed PVB grades from automotive and construction sector. The plasticizer content and their chemical nature also determined authors.
The manuscript is well written and well supported by the data. Only few minor things should be corrected before the acceptance of the paper.
For example, Eq1 should be in high resolution.
The inset images of Figure 8 is not clear, please make them more big and high resolution. Same for Figure 9b also.
Point 1: Eq1 should be in high resolution.
Response: Corrected. It was retyped with higher resolution and highlighted in the text.
Point 2: The inset images of Figure 8 is not clear, please make them more big and high resolution. Same for Figure 9b also.
Response: Corrected. The inset images were maximized and high resolution was used.

Reviewer 4 Report
Comments and Suggestions for Authors
This paper presents a straightforward experimental analysis of PVB. It combines several methods and provides useful information on a polymer seldom reported in the literature. I support publication pending some minor corrections:
Several acronyms are not defined. Do it the first time they appear.
Revise for sub/superscripts everywhere.
Several spaces are missing throughout the manuscript. Revise the whole document.
Line 236: change “ascribable” by “related”.
Line 291 ; change “diagnostic” by “specific”.
Table 2: several SD are reported as 0.0, which is highly unlikely…
Line 418 ; change “received” by “residual”.
Line 445 and 446 ; change “receive” by “produce”.
Table 6. Put the title above.
References: write the journals’ names uniformly and according to the format required.
Figure S1 is awful !!! Remove the Supplementary Material.
Comments on the Quality of English Languageminor corrections as described above
Author Response
Reviewer 4
This paper presents a straightforward experimental analysis of PVB. It combines several methods and provides useful information on a polymer seldom reported in the literature. I support publication pending some minor corrections:
Several acronyms are not defined. Do it the first time they appear.
Revise for sub/superscripts everywhere.
Several spaces are missing throughout the manuscript. Revise the whole document.
Line 236: change “ascribable” by “related”.
Line 291 ; change “diagnostic” by “specific”.
Table 2: several SD are reported as 0.0, which is highly unlikely…
Line 418 ; change “received” by “residual”.
Line 445 and 446 ; change “receive” by “produce”.
Table 6. Put the title above.
References: write the journals’ names uniformly and according to the format required.
Figure S1 is awful !!! Remove the Supplementary Material.
Point 1: Several acronyms are not defined. Do it the first time they appear.
Response: Corrected. MeOD and CPMAS were defined as well as all the used methods.
Point 2: Revise for sub/superscripts everywhere.
Response: Done. Highlighted in yellow in the revised version of the manuscript.
Point 3: Several spaces are missing throughout the manuscript. Revise the whole document.
Response: Corrected. Several missing spaces were found in the names of authors. Also, in Fig.1 caption.
Point 4: Line 236: change “ascribable” by “related”.
Response: Corrected and highlighted in yellow in the revised version of the manuscript
Point 5: Line 291; change “diagnostic” by “specific”.
Response: Corrected and highlighted in yellow in the revised version of the manuscript.
Point 6: Table 2: several SD are reported as 0.0, which is highly unlikely…
Response: Corrected. In these cases, SD was lower than <0.1% thus two decimal digits would be required (for example 0.04 was rounded at 0.0). It is now corrected as 0.1% to be more realistic.
Point 7: Line 418; change “received” by “residual”.
Response: Corrected and highlighted in yellow in the revised version of the manuscript.
Point 8: Line 445 and 446; change “receive” by “produce”.
Response: Corrected and highlighted in yellow in the revised version of the manuscript.
Point 9: Table 6. Put the title above.
Response: Corrected and highlighted in yellow in the revised version of the manuscript.
Point 10: References: write the journals’ names uniformly and according to the format required.
Response: Corrected. All the Journal names were abbreviated as it is required by the format. We updates references ‘ Ref. 12 Jordan J. Mech. Ind. Eng’ and Ref. 15 ‘Int. j. sci. res’.
Point 11: Figure S1 is awful !!! Remove the Supplementary Material.
Response: Figure S1 was removed.
